# DeepCircuitX: Repository-Level RTL Dataset for Code Understanding, Generation, and Multimodal Analysis

## Abstract

This paper introduces **DeepCircuitX**, a comprehensive multimodal dataset designed to advance RTL code understanding, generation, and completion tasks in hardware design automation. Unlike existing datasets, which focus either on file-level RTL code or downstream netlist and layout data, DeepCircuitX spans repository, file, module, and block-level RTL code, providing a more holistic resource for training and evaluating large language models (LLMs). The dataset is enriched with Chain of Thought (CoT) annotations that offer detailed functionality and structure descriptions at multiple levels, enhancing its utility for RTL code understanding, generation, and completion.

In addition to RTL data, DeepCircuitX includes synthesized netlists and power-performance-area (PPA) metrics, allowing for early-stage design exploration and PPA prediction directly from RTL code. We establish comprehensive benchmarks for RTL code understanding, generation, and completion using open-source models such as CodeLlama, CodeT5+, and CodeGen, demonstrating substantial improvements in task performance. Furthermore, we introduce and evaluate models for PPA prediction, setting new benchmarks for RTL-to-PPA analysis. We conduct human evaluations and reviews to confirm the high quality and functionality of the generated RTL code and annotations. Our experimental results show that DeepCircuitX significantly improves model performance across multiple benchmarks, underscoring its value as a critical resource for advancing RTL code tasks in hardware design automation.

## 1 Introduction

Register Transfer Level (RTL) modeling stands as a pivotal and early step in the Electronic Design Automation (EDA) flow. RTL code, such as Verilog and VHDL, acts as a high-level abstraction that represents the functionality of hardward designs, bridging between design specifications and circuit implementations. Following RTL modeling, engineers across various departments in a semiconductor companies translate RTL code into netlists, floorplans and layouts. Consequently, RTL representation significantly impacts the quality of circuit designs.

To enhance the current EDA tools for complex modern chip design and expedite time-to-market, integration of Artificial Intelligence (AI) techniques into the EDA flow, particularly for RTL-related tasks, has emerged as a promising avenue. For instance, Allam & Shalan (2024) employs large language models (LLMs) to understand and describe RTL designs in natural language to assist engineers. Cui et al. (2024); Thakur et al. (2023) focus on generating RTL code from the design specification. Clearly, the effectiveness of deep learning models is heavily contingent upon the quality of the training data (Chang et al., 2024). However, we observe that existing RTL datasets suffer from notable limitations, hindering the application of AI-based solutions in practical RTL modeling and verification.

One prominent limitation is the narrow scope of most datasets. Since the semiconductor ecosystem is far inferior to software openness, the accessible designs are limited. Many datasets either focus on a limited set of circuit types, often centering around a single design such as processors (Chai et al., 2022; Jiang et al., 2024)), or indiscriminately collect all Verilog files without accounting for their

correctness and distribution (Thakur et al., 2023; Wu et al., 2024). This restricted diversity hampers the generalizability and effectiveness of these datasets.

Another critical limitation is the lack of focus on circuit implementation in RTL-based datasets. For instance, RTLLM (Lu et al., 2024) gathers RTL specifications and verifies the functional correctness of RTL code, while Chang et al. (2024) ensures functionality when constructing and augmenting datasets. However, these works do not consider or include circuit-level information, such as netlists and layouts, which are crucial in later stages of the EDA flow. As a result, these datasets overlook the correlation between RTL code and its corresponding circuit implementations. Consequently, training RTL models with the existing datasets that both guarantee functional correctness and optimize Power, Performance, and Area (PPA) remains a significant challenge.

To overcome these limitations, we present a multimodal circuit dataset for deep learning in EDA. Our dataset is meticulously curated according to the scope of the real-world chip and collect more than 4,000 circuit design projects from various data sources. Unlike previous datasets that provide verilog files only, each data point in our dataset is structured as a complete repository and also splitted into files, modules and blocks for various scenarios. This multi-tiered structure enables training models at different scales, making it suitable for a variety of LLMs and models. Moreover, the repository-level data allows logic synthesis and physical design flows to convert RTL code into various circuit modalities, including Control/Data Flow Graph (CDFG) of RTL code, And-Invertor Graph (AIG), post-mapping netlist, floorplaning and layout. Therefore, we can build the interactions between RTL designs and circuit implementations.

Furthermore, we meticulously label the circuits for unimodal RTL tasks and cross-modal PPA prediction to enhance accessibility of our dataset. We propose a Chain of Thought (CoT)(Wei et al., 2022) detailed annotation method to generate descriptions and comments for each of the four levels, namely, repo-level, file-level, module-level and block-level. By using GPT-4(Achiam et al., 2023) and Claude(Anthropic, 2024), we leverage annotations from higher levels to assist in annotating lower levels. Additionally, we generate question-answer pairs to help describe the functionality and key features of each code segment, enabling better training data for LLMs. Moreover, we provide the corresponding logic synthesis results for the cross-stage PPA prediction on RTL code.

With this dataset, we can construct a variety of pre-training tasks, evaluation tasks, and benchmarks, enabling the study of LLM performance on RTL code understanding and completion. Our experiments include tasks that assess LLMs' abilities in RTL code comprehension, generation and completion, we trained various models, including CodeLLama(Roziere et al., 2023), CodeT5+(Wang et al., 2023), CodeGen(Nijkamp et al., 2022), and DeepSeek(Liu et al., 2024; Zhu et al., 2024), across different scales ranging from 220M to 16B on our dataset. The results demonstrate two key findings: 1) Every large model fine-tuned on our dataset significantly outperforms its original, non-fine-tuned counterpart across all metrics, highlighting the effectiveness of our data. 2) LLMs of different scales, such as the 220M CodeT5, 7B and 16B models, show substantial improvements after fine-tuning, reflecting the adaptability and generalization capabilities of our dataset across varying model sizes. Moreover, to prove the contribution of our dataset on the EDA tasks, we utilize learning-based PPA prediction models, the results show that accurate predicting PPA on early stage still poses a question, providing valuable insights for future research in RTL and hardware design automation. And the pipeline and framework are illustrated in Figure 1.

The main contributions of our work are summarized as follows:

- We propose a holistic dataset including over 4,000 repository-level RTL projects, covering chip-level, IP-level, module-level and RISCV designs, and incorporating a diverse range of functional and algorithmic keywords, as well as High-Level Synthesis (HLS) data converted to RTL.

- Our dataset is organized into four levels, namely, repository, file, module, and block levels, allowing models to be trained at different scales and enabling broader applications in EDA tasks such as synthesis, netlist generation, PPA analysis, and layout design.

- We propose a Chain of Thought (CoT) annotation method using GPT-4 and Claude to generate detailed comments, descriptions, and question-answer pairs, improving training data for LLMs in RTL code understanding and generation.

- We create pre-training and evaluation benchmarks for LLMs on RTL tasks such as code understanding, completion, and neural network-based PPA prediction, demonstrating the effectiveness, adaptability and generalization capabilities of our dataset for both RTL code comprehension and EDA tasks.

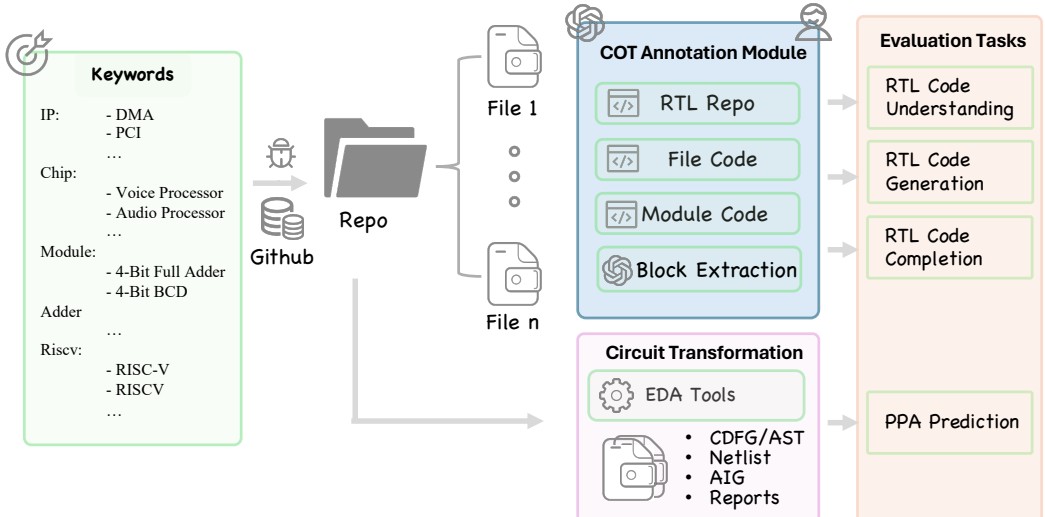

Figure 1: Pipeline overview of the proposed framework, illustrating the key stages: data collection from GitHub using keywords, data annotation via chain-of-thought (COT), circuit transformation, and evaluation, including RTL code tasks for LLM and PPA prediction.

## 2 RELATED WORK

### 2.1 PREVIOUS DATASET IN EDA

AI-based methodologies excel in addressing classification, prediction, and optimization tasks, making them well-suited for chip design. Over the past decade, the integration of AI into Electronic Design Automation (EDA) has emerged as an attractive direction in the fields of chip design and semiconductor industry, which is well surveyed in Huang et al. (2021); Chen et al. (2024).

The cornerstone of training AI models lies in the quality of the dataset utilized. In the domain of EDA, existing datasets can be broadly categorized into two classes. Firstly, there are unimodal datasets or benchmarks such as ISCAS'89 (Brglez et al., 1989), ITC'99 (Davidson, 1999), IWLS'05 (Albrecht, 2005) and EPFL (Amarú et al., 2015) benchmarks, which offer open-source circuit netlists primarily for front-end applications like logic synthesis and design for test. Shrestha et al. (2024) presents a dataset of physical designs generated from the IWLS'05 benchmark circuit suite, utilizing the open-source 130nm Process Design Kit (PDK) by Skywater and the OpenROAD toolkit (Ajayi & Blaauw, 2019). Despite these efforts, these datasets still lack in providing comprehensive insights into cross-stage EDA tasks, such as optimizing circuit designs at the front-end for improved PPA metrics during the back-end stage.

Secondly, other multimodal circuit datasets are generated using EDA tools. For instance, Circuit-Net (Chai et al., 2022; Jiang et al., 2024) creates a dataset through logic synthesis and physical design, where the circuit designs are synthesized into gate-level netlists and transformed into layouts using commercial EDA tools. Jiang et al. (2024) expand this work by providing data for million-gate designs such as CPUs, GPUs, and AI chips, and using the 14nm FinFET technology node to capture the increased complexity of manufacturing and modeling. Nonetheless, the existing datasets encompass a limited range of circuit designs, thereby constraining the generalizability of subsequent model training efforts.

## 2.2 RTL-stage Understanding, Completion, and Generation

Thakur et al. (2023) collects approximately 50,000 open-source Verilog code samples and fine-tunes five pre-trained LLMs, with model sizes ranging from 345 million to 16 billion parameters. Liu et al. (2023) introduces a comprehensive evaluation dataset consisting of 156 problems sourced from HDLBits and developed a benchmarking framework to automatically test the functional correctness of Verilog code completions. Similarly, Thakur et al. (2024) fine-tunes existing LLMs on Verilog datasets collected from GitHub and textbooks, evaluating the functional correctness of the generated code using a custom test suite. Chang et al. (2024) designs a data augmentation framework for training Chip Design LLMs, enabling them to generate Verilog code, EDA scripts, and coordinate EDA workflows based on natural language design descriptions. They also benchmark their approach by fine-tuning Llama 2 models with 7 billion and 13 billion parameters. Lastly, Zhang et al. (2024) presents the MG-Verilog dataset, an open-source dataset that meets essential criteria for high-quality hardware data, facilitating the effective use of LLMs in hardware design. However, these datasets focus solely on file-level RTL code, neglecting the comprehensive information contained within entire RTL project designs, modules, and code blocks. As a result, they fail to provide the multimodal data, such as netlists and PPA metrics, that can be obtained through synthesis. Moreover, existing studies primarily focus on RTL code generation and completion, often overlooking the critical aspects of annotations, comments, and descriptions for RTL code. Hence, there has been little progress in advancing RTL code understanding tasks, particularly for LLMs.

# 3 Methodology

## 3.1 Data Preparation

Our dataset distinguishes itself by focusing on chip-level, IP-level, module-level RTL designs, and RISC-V architectures. We compile a list of 222 keywords representing these levels from sources like Alldatasheet (Alldatasheet, 2003) and other relevant websites. Examples include chip-level designs such as voice processors, audio processors, and video processors; IP-level designs like DMA, PCI, and true random number generators; and module-level designs including 4-bit binary full adders, arithmetic logic units, and multiplier-accumulators. We collect over 4,000 repository-level RTL projects, which encompass 140,000 RTL files across 77 functional categories, as shown in Table 1. Moreover, the detailed data cases are illustrated in Figure 2.

To construct the RTL-language dataset, we organize the data into four distinct levels: repository, file, module, and block. We employ a Chain of Thought (CoT) approach for RTL code annotation, leveraging GPT-4 (Achiam et al., 2023) and Claude (Anthropic, 2024) to generate detailed comments, descriptions, and question-answer pairs. This methodology enhances the training data for large language models (LLMs) in RTL code understanding and generation, with further details provided in Section 3.2. Additionally, to develop the multimodal dataset, we synthesize the RTL projects to obtain netlists, power, performance, and area (PPA) metrics, as well as layout designs. Further details will be discussed in Section 3.3.2.

Table 1: Summary of data across different levels in DeepCircuitX. The table presents the number of function categories, repositories, and RTL files at the Chip, IP, Module, and RISC-V levels.

| Level | Function Categories | Repo Number | RTL File Number |
|---|---|---|---|
| Chip Level | 17 | 109 | 5508 |
| IP Level | 3 | 225 | 12961 |
| Module Level | 57 | 2383 | 38692 |
| RISCV | - | 2078 | 98450 |

## 3.2 LANGUAGE-RTL CODE DATASET CONSTRUCTION

### 3.2.1 CHAIN OF THOUGHT (CoT) FOR RTL CODE ANNOTATION

The Chain of Thought (CoT) reasoning method, commonly used in deep learning and natural language processing, simulates human reasoning by guiding models through structured steps. We adopt this approach to create detailed annotations for Verilog code, ensuring a comprehensive understanding at different levels of the RTL design. The annotation process is carried out at three distinct levels: *module-level*, *block-level*, and *repository-level*. The detailed pipeline is shown in Figure 4 in the Appendix.

**Module-Level Annotations** To annotate RTL modules, we utilize a multi-round question-answering approach to extract key information from the Verilog code, focusing on aspects such as the module's name, input/output ports, and internal signals. This process begins with structured questions designed to clarify both what the module does (What) and how it achieves this functionality (How). The responses provide the foundational details about the module's components and behavior. Using this information, a detailed specification is created that summarizes the module's name, its purpose, the roles of the input and output ports, and an explanation of the internal signals. Additionally, the specification includes a breakdown of the functional blocks within the module. Finally, with this specification and the original code, ChatGPT generates a concise yet comprehensive module-level annotation, encapsulating both the functionality and the implementation details.

**Block-Level Annotations** Block-level annotations offer detailed insights into the functionality of specific sections within an RTL module. Given the complexity of Verilog code, which often includes nested structures, segmenting it using regular expressions alone can be challenging. To address this, we leverage GPT-4 for handling these intricacies. The code is divided into distinct functional blocks, including constructs such as `always`, `initial`, `task`, `function`, `generate`, `assign`, and `final` blocks. While regular expressions are employed to improve segmentation accuracy for straightforward patterns, GPT-4 is crucial for managing more complex structures. Once the blocks are identified, each is annotated to describe both its purpose and how it functions, ensuring that the role of each block within the module is clearly defined, thereby enhancing the overall understanding of the module's functionality.

**Repo-Level Annotations** At the repository level, we aim to provide an overarching understanding of the entire RTL project. This includes information about the structure, purpose, and interconnections of various modules and files within the repository. To achieve this, we gather the file structure information and combine it with module-level annotations to form a complete picture of the project. GPT-4 is then used to summarize the contents of the repository, producing a top-down view of how individual files and modules work together. This approach allows the repo-level annotation to reflect both functional and design-level details, capturing the broader goals of the RTL project.

Table 2: Summary of annotated RTL categories at the module, block, and repository levels in Deep-CircuitX. The table highlights the number of annotated modules, blocks, and repositories for each RTL category, including Chip, IP, Module, and RISC-V.

| RTL categories | Module-Level | Block-Level | Repo-Level |
|---|---|---|---|
| Chip | 5471 | 36955 | 84 |
| IP | 12863 | 20101 | 183 |
| Module | 28901 | - | 1389 |
| RISCV | 2116 | - | 560 |

### 3.2.2 DATASET FOR RTL CODE UNDERSTANDING, COMPLETION AND GENERATION

In addition to annotations, we constructed a dataset that supports three distinct tasks related to RTL code: *understanding*, *completion*, and *generation*. Table 3 shows the data counts across RTL categories (IP, Module, RISC-V, and Chip) for each task.

Table 3: Data counts for code generation, code completion, and comment generation tasks across different categories.

| Tasks Dataset | IP | Module | RISC-V | Chip | All |
|---|---|---|---|---|---|
| RTL Code Understanding | 6386 | 14499 | 1348 | 3922 | 26155 |
| RTL Code Completion | 6178 | 14131 | 1312 | 3822 | 25443 |
| RTL Code Generation | 6479 | 16511 | 1393 | 3950 | 28333 |

**RTL Code Understanding**    This task evaluates the model's ability to interpret and describe RTL code. Given a module's RTL code as input, the model generates a detailed, concise description, covering key aspects such as the module's purpose, input/output signals, internal logic, and overall behavior. This task is crucial for assessing the model's ability to generate human-readable explanations for code analysis and documentation.

**RTL Code Completion**    In this task, the model is provided with a partial RTL code (typically the module header with input/output ports and parameters). The goal is for the model to complete the code by generating the missing internal logic, control structures, and signal definitions. This task mirrors autocompletion functionality found in modern code editors and evaluates the model's ability to infer and generate code from context.

**RTL Code Generation**    In the RTL code generation task, the model is tasked with producing a full implementation of RTL code based on a high-level description and specified input and output parameters. The goal is to generate a fully functional Verilog module that adheres to the provided specifications. This task assesses the model's ability to translate design requirements into precise RTL implementations, which is critical for automating the hardware design process.

### 3.3 Multimodal Transformation of RTL Code

#### 3.3.1 Graph-based Code Representation

Abstract Syntax Tree (AST) serves as a tree-based representation to model the structure of programming code, extensively used in syntax analysis and compilation. In the context of Verilog code, each node within the AST denotes a variable or operator, and the edges formulate the relations between these nodes. Additionally, Control/Data Flow Graph (CDFG) (Orailoglu & Gajski, 1986) is a another graph-based code representation to visualize how data moves between registers and how control signals dictate the operation of the design. CDFG proves advantages for modeling and verifying the functionality of RTL designs (Coussy et al., 2009; Vasudevan et al., 2021). We generate the AST and CDFG of RTL designs with open-source tools Yosys (Wolf, 2016) and customized Pyverilog (Takamaeda-Yamazaki, 2015).

#### 3.3.2 Circuit Netlist Synthesis

Our dataset contains RTL repositories with module invocations and complete Verilog files, enabling us to derive the circuit netlists via logic synthesis process. For each RTL repository, we employ the commerical tool Synopsys Design Compiler 2019.12 to transform HDL code into netlists. The RTL designs are mapped into several open-source technology libraries, including GlobalFoundries 180nm, skywater 130nm, ihp-sg 130nm, nangate 45nm and asap 7nm. Each RTL file is synthesized using both the compile and *compile_ultra* commands. By toggling the set max area 0 option, we obtain both the default netlist and a netlist optimized for maximum area constraints. We assess the internal, transition, and leakage power of each mapped netlist using PrimeTime (Synopsys, 2023.12). The critical path delay is reported by the Design Compiler.

Finally, we store the post-mapping netlist into both verilog format (.v files) and Standard Delay Format (SDF) (Sagdeo, 1998) (.sdf files). Moreover, we record the logic synthesis reports (.rpt files) with maximum path delay, area and dynamic power for the following experiments. We also convert the post-mapping netlist into And-Inverter Graph (AIG) format, a prevalent and widely-adopted common format for netlist learning (Li et al., 2022; Shi et al., 2023; 2024; Deng et al., 2024), using

abc (Brayton & Mishchenko, 2010) tool for further investigation. The multi-modal data cases are illustrated in Figure 3.

# 4 BENCHMARK AND EXPERIMENTS

## 4.1 OVERVIEW

To evaluate our multi-modal dataset DeepCircuitX and establish benchmarks, we design experiments of human evaluation in Section 4.2, RTL code tasks in Section 4.3 and Section 4.4, and PPA prediction experiments in Section 4.5.

Specially, in Section 4.3 and Section 4.4, we establish benchmarks for LLMs tailored to RTL tasks, utilizing open-source models such as CodeLlama (Roziere et al., 2023), CodeT5+ (Wang et al., 2023), deepseek-coder (Guo et al., 2024) and CodeGen (Nijkamp et al., 2022). Our annotated data is employed to fine-tune these models for RTL code understanding, completion, and generation tasks. Here is an concise iintroduction of the based-LLMs we select:

- CodeLlama is fine-tuned Llama on a diverse corpus of code from various programming languages, enabling it to understand syntax, semantics, and the context of code snippets.
- CodeT5+ is an enhanced version of the original CodeT5 model, specifically optimized for code understanding and generation tasks. Building on the strengths of its predecessor.
- CodeGen is trained on GPT-3 (Floridi & Chiriatti, 2020) by a diverse range of programming languages and employs advanced techniques to understand and generate code efficiently.
- DeepSeek-V2-lite (Liu et al., 2024) has broader applicability on general tasks of natural language and DeepSeek-Coder-V2-lite (Zhu et al., 2024) focuses on code-related tasks, offering fine-tuned capabilities for high-quality code generation and synthesis.

## 4.2 HUMAN EVALUATION

To evaluate the effectiveness of our generating comments approach and the quality of the annotations, we conduct a series of evaluation criteria and metrics involving human reviews by independent experienced engineers. We employ the following metrics: Accuracy, Completeness and Understandable Clarity. The detailed grading criteria is introduced in the Appendix A.

Engineers are tasked with analyzing both the RTL code and the accompanying annotations based on specific criteria, ensuring a thorough evaluation of the provided information. Subsequently, they assign a grade to each metric on a scale from 1 to 4, with 4 indicating the highest quality and 1 indicating the lowest. To ensure fairness, each generated text is reviewed by 5 individuals, and the average score is recorded in Table 4. We observe that the code annotations in DeepCircuitX exhibit high quality, with all metrics scoring above 3.5 out of 4.

Table 4: Human evaluation grading of repo-level annotation and module-level annotation, we use the metrics of accuracy, completeness, understandable clarity to evaluate the quality of our annoation.

| Metrics | Repo Annotation | Module Annotaion |
|---|---|---|
| Accuracy | 3.74/4 | 3.5/4 |
| Completeness | 3.79/4 | 3.78/4 |
| Understandable Clarity | 3.84/4 | 3.76/4 |

## 4.3 RTL CODE UNDERSTANDING

### 4.3.1 EVALUATION METRICS

**BLEU** measures n-gram overlap (1 to 4) between generated text and reference translations, incorporating a brevity penalty. **METEOR** accounts for synonymy and stemming, calculating a harmonic mean of precision and recall. **ROUGE** focuses on summarization, with variants like ROUGE-N for

n-gram overlap and ROUGE-L for the longest common subsequence, emphasizing recall to assess content relevance.

### 4.3.2 Analysis

The experimental results, shown in Table 5, demonstrate the performance of various base Large Language Models (LLMs) and fine-tuned LLMs on our RTL code understanding benchmark, using evaluation metrics BLEU-4, METEOR, ROUGE-1, ROUGE-2, and ROUGE-L.

Initially, the original versions of the LLMs, such as CodeLLama, CodeT5+, CodeGen2, and DeepSeek, exhibit relatively low performance across most metrics. For example, CodeLLama (original) achieves a BLEU-4 score of 0.0828, while CodeGen2.5 (original) shows moderate improvement with a BLEU-4 score of 0.1060. Among the original models, DeepSeek-Coder-V2-lite (original) stands out, significantly outperforming others with a BLEU-4 score of 2.2387 and a ROUGE-1 score of 30.3208, indicating its strong baseline performance even before fine-tuning.

After fine-tuning on our dataset, every large model demonstrates significantly better performance across BLEU-4, METEOR, ROUGE-1, ROUGE-2, and ROUGE-L metrics compared to their original, non-fine-tuned counterparts. This highlights the effectiveness of our dataset. Moreover, models of various sizes, such as the 220M CodeT5, as well as larger 7B and 16B models, all show substantial improvements after fine-tuning. This indicates that our dataset is well-suited for models of different scales, providing strong adaptability and generalization.

Overall, the fine-tuned LLMs significantly outperform their original counterparts, indicating the importance of fine-tuning on domain-specific datasets like ours.

Table 5: The results of our base-LLMs and fine-tuned LLMs by our training dataset on our RTL code understanding benchmark.

| Based LLM | BLEU-4 | METEOR | ROUGE-1 | ROUGE-2 | ROUGE-L |
|---|---|---|---|---|---|
| CodeLLama (original) | 0.0828 | 5.9414 | 11.0046 | 0.3139 | 0.2581 |
| CodeT5+ 220m-bimoal (original) | 0.1410 | 4.3277 | 10.9925 | 0.5076 | 9.3043 |
| CodeGen2 (original) | 0.1082 | 3.7890 | 8.0311 | 0.1173 | 7.2161 |
| CodeGen2.5 (original) | 0.1060 | 4.8271 | 9.4001 | 0.3698 | 8.5856 |
| DeepSeek-Coder-V2-lite (original) | 2.2387 | 24.3311 | 30.3208 | 6.3163 | 27.4282 |
| DeepSeek-V2-lite (original) | 0.2311 | 3.3951 | 7.3720 | 0.2646 | 6.3360 |
| CodeLLama (7b) | 0.8619 | 18.2621 | 25.0461 | 6.4493 | 22.8182 |
| CodeT5+ (220m-bimodal) | 4.9067 | 23.5043 | 34.8671 | 9.9023 | 32.1642 |
| CodeGen2 (1b) | 7.7605 | 27.8049 | 37.2150 | 13.1385 | 34.0647 |
| CodeGen2.5 (7b) | 13.6858 | 34.7494 | 43.5244 | 18.5249 | 40.2223 |
| DeepSeek-Coder-V2-lite (16b) | 11.9180 | 33.5011 | 41.8527 | 17.2014 | 38.0473 |
| DeepSeek-V2-lite (16b) | 13.6972 | 39.5962 | 43.3732 | 19.0589 | 39.5962 |

### 4.4 RTL Code Completion and Generation

### 4.4.1 Evaluation Metrics

In the realm of RTL code completion and generation, the evaluation of model performance is critical to advancing intelligent programming tools. The Pass@k metric serves as a pivotal measure in this domain, quantifying the accuracy of code generation models by assessing their ability to produce valid solutions within the top-k predictions. Specifically, Pass@k evaluates whether the correct code snippet appears among the model's top k outputs, thereby providing insights into both the effectiveness and reliability of the model's predictions.

### 4.4.2 Analysis

Table 6 compares the performance of both original and fine-tuned LLMs on RTL code completion and generation tasks, focusing on Pass@1 and Pass@5. Notably, every model fine-tuned with our

dataset significantly outperforms its original, non-fine-tuned counterpart, demonstrating the effectiveness of our data. Additionally, for models of different scales, such as the 220M CodeT5 and 7B models, the results after fine-tuning show substantial improvements. This highlights the adaptability and generalization capability of our dataset across various model sizes.

Table 6: Pass@K results for RTL code completion and generation across various LLMs.

| Based LLM | Pass@1 | | Pass@5 | |
|---|---|---|---|---|
| | RTLLM | VerilogEval | RTLLM | VerilogEval |
| CodeLLama (original) | 0 | 0.64% | 0 | 0 |
| CodeT5+ 220m-bimodal (original) | 0 | 0 | 0 | 0 |
| CodeGen2 (original) | 0 | 0 | 0 | 0.64% |
| CodeGen2.5 (original) | 17.24% | 23.08% | 17.24% | 24.36% |
| CodeLLama (7b) | 6.90% | 1.92% | 6.90% | 6.41% |
| CodeT5+ (220m-bimodal) | 3.45% | 4.49% | 3.45% | 4.49% |
| CodeGen2 (1b) | 13.79% | 10.90% | 13.79% | 10.90% |
| CodeGen2.5 (7b) | 24.14% | 24.36% | 24.14% | 25% |

## 4.5 PPA Prediction

### 4.5.1 Overview

Optimizing Power, Performance, and Area (PPA) stands out as a primary objective in the circuit design process. Estimating PPA metrics at an early stage not only boosts design efficiency but also empowers a more agile response to evolving design requirements and constraints. Unlike previous RTL datasets that solely gather Verilog files, our datasets encompass entire repositories, allowing us to obtain post-mapping netlists and logic synthesis reports by EDA tools. Consequently, we can parse the PPA metrics from the logic synthesis reports and predict them for the corresponding RTL designs at an early stage. In this subsection, we formulate the PPA prediction task within our dataset and assess the effectiveness of current learning-based prediction models (Xu et al., 2022; Sengupta et al., 2022; Fang et al., 2023).

### 4.5.2 Evaluation Metrics

We regard the circuit charateristics of the post-mapping netlist produced by the Design Compiler tool with default settings as the ground truth for PPA metrics. Specifically, we define the dynamic power under random workloads as the **Power** metric, the maximum path delay as the **Delay** metric and the total cell area as the **Area** metric.

We use Mean Absolute Error Percentage (MAPE) and Root Relative Square Error (RRSE) to evaluate the prediction accuracy of PPA. It should be denoted that the PPA predictor trained on the datasets generated with a certain technology library cannot be generalized to another, as the PPA metrics of netlists are strongly linked to the technology library. Therefore, we choose the netlists and logic synthesis reports with a specific library, which keeps consistency with the settings in Xu et al. (2022); Sengupta et al. (2022); Fang et al. (2023). We opt for skywater 130nm library in the following experiments.

### 4.5.3 Analysis

We collect 146 RTL designs for training and 10 designs for testing, with these designs containing more than 10k cells after logic synthesis and closely resembling practical designs. To explore the data scalability of PPA prediction models, we sample the training dataset size to 10%, 50% and 100%. It should be denoted that we exclude the model (Sengupta et al., 2022) for the delay prediction as it cannot support this task. According to the prediction results shown in Table 7, we conclude three observations. Firstly, the accuracy of predictions improves as the training data volume increases. For instance, the MAPE of area prediction is 4.3201 with 10% data and decreases to 0.3303 with 100% data. Secondly, PPA predictors demonstrate weaker performance in delay prediction.

Table 7: PPA prediction results of learning-based models (the lower, the better).

| Model | Data | Area | | Power | | Delay | |
|---|---|---|---|---|---|---|---|
| | | MAPE | RRSE | MAPE | RRSE | MAPE | RRSE |
| Xu et al. (2022) | 10% | 1.9379 | 1.4913 | 1.9045 | 1.1362 | 49.3638 | 6.6901 |
| | 50% | 0.7811 | 1.9041 | 0.8425 | 1.1759 | 47.7314 | 2.8816 |
| | 100% | 0.6564 | 1.7197 | 0.7549 | 1.1740 | 4.7392 | **2.3647** |
| Sengupta et al. (2022) | 10% | 1.9066 | 1.0802 | 10.2783 | 14.7510 | - | - |
| | 50% | 0.7954 | 0.6133 | 0.7478 | **0.3539** | - | - |
| | 100% | 0.5996 | **0.5074** | 0.7343 | 0.6558 | - | - |
| Fang et al. (2023) | 10% | 1.3405 | 1.1197 | 2.3253 | 29.7767 | 80.7779 | 8.5936 |
| | 50% | 0.5640 | 0.9982 | 1.7237 | 29.8749 | 24.7716 | 6.8725 |
| | 100% | **0.3303** | 0.7605 | **0.6501** | 2.2541 | **3.4769** | 4.2863 |

For example, both models exhibit unsatisfactory performance, where (Xu et al., 2022) shows 4.7392 MAPE and (Fang et al., 2023) has 3.4769 MAPE. This suggests that estimating timing characteristics in the early stages using path features on RTL-based graphs is still difficult, as logic synthesis tools heavily optimize logic to minimize maximum path delays. Thirdly, in comparison to the originally reported performance on simple benchmarks, these models exhibit diminished performance on designs more than 10k cells in our dataset. Therefore, how to accurately predict PPA of practical designs remains an opening question, necessitating further exploration in the EDA community.

## 5 LIMITATION

One key limitation of DeepCircuitX is that it emphasizes the diversity of RTL project categories, but does not focus as much on the quantity of RTL code. Additionally, due to time constraints and the speed of annotation, we selected only high-quality data for annotation and processing, including synthesis for AST, CDFG, PPA, and Netlist generation. Furthermore, we have not yet conducted specific experiments to evaluate the quality of the generated AST and CDFG representations. In future iterations of DeepCircuitX, we plan to conduct comprehensive experiments to assess the quality and performance of the AST, CDFG, and Netlist components.

## 6 CONCLUSION

In this paper, we introduce **DeepCircuitX**, a comprehensive and multimodal dataset tailored to advancing RTL code understanding, generation, and completion tasks in hardware design automation. By offering a holistic resource that spans repository, file, module, and block-level RTL code, DeepCircuitX enables large language models to better tackle the complexities of hardware design. The integration of Chain of Thought (CoT) annotations further enhances the dataset's value by providing detailed insights into functionality and structure, thereby improving model training and performance across a variety of RTL tasks. Our experiments demonstrate that models trained on DeepCircuitX significantly outperform existing methods in tasks like code understanding, generation, and completion, as well as in RTL-to-PPA prediction. The inclusion of synthesized netlists and PPA metrics opens up new avenues for early-stage design exploration, offering a practical tool for both researchers and practitioners in electronic design automation (EDA). As a result, DeepCircuitX establishes new benchmarks for RTL tasks, setting a foundation for future innovations in hardware design automation and demonstrating the potential of LLMs to transform this critical domain.

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

## A    HUMAN EVALUATION GRADING CRITERIA

1. The grading criteria of **Accuracy**

- 4: Completely Accurate: The annotation accurately captures all functionalities, I/O signals, and operational logic without errors.
- 3: Partially Accurate: The annotation contains some inaccuracies in a limited number of functionalities, I/O signals, or operational logic descriptions.
- 2: Not Quite Accurate: The annotation has significant inaccuracies affecting a majority of functionalities, I/O signals, or operational logic descriptions.
- 1: Completely Incorrect: The annotation fails to accurately describe the functionalities, I/O signals, or operational logic.

2. The grading criteria of **Completeness**

- 4: Fully Complete: The annotation provides a comprehensive explanation of overall functionality, I/O signals, and operational logic.
- 3: Mostly Complete: The annotation is largely complete but misses minor details regarding overall functionality, I/O signals, or operational logic.
- 2: Not Quite Complete: The annotation is partially complete but misses many details regarding overall functionality, I/O signals, or operational logic.
- 1: Incomplete: The annotation lacks significant portions of the overall functionality, I/O signals, or operational logic.

3. The grading criteria of **Understandable Clarity**

- 4: Clear and Concise: The annotation is articulated clearly with minimal redundancy.
- 3: Relatively Clear: The annotation is generally clear but contains some redundant information.
- 2: Vague: The annotation is unclear and lacks precision.
- 1: Inaccurate or Incomplete: The annotation is both inaccurate and incomplete.

## B    CASE: ANNOTATION

As illustrated in Figure 2, our dataset preserves the original file structure of the repository while extracting repository-level annotations and providing multi-level annotations for each Verilog file in the original repository. For each Verilog file, all annotation information is stored in a folder named "*<original file name>*", which contains:

- `<original file name>.txt`: A text file containing the module-level annotations;
- `<original file name>.v`: The original Verilog file;
- `blocks`: A folder that contains the original code for each block from the Verilog file (`block{i}.v`) along with the corresponding block-level annotations (`block{i}.txt`). Each annotation includes the line numbers indicating the block's position in the original code and a comment describing its functionality;
- `intermediate_comment`: A folder containing intermediate results generated by the LLM during the Chain of Thought (CoT) process;
- `spec`: A folder containing a file named `spec.txt`, which provides a detailed specification of the module.

When we construct the agent of module-level RTL code annotation, each round focuses on different aspects of the code:

**Round 1:**

- What is the name of the module in the code?
- Provide a brief explanation of the module's functionality.

**Round 2:**

- What are the input and output ports of the module?
- Provide a simple explanation of each input and output port, and their respective roles within the module.

**Round 3:**

- What are the internal signals used in the module?
- Provide a brief explanation of each internal signal and its role in the module.

## C    CASE: CIRCUIT TRANSFORMATION

Figure 3 shows an example of our dataset after Circuit transformation, including the snippet of RTL design, are report and the snippet of post-mapping netlist. The file structures are listed as follows:

- `filelist.f`: The list of all verilog used for logic synthesis.
- `graph`: The directory of graph-based circuit representations, including the CDFG `cdfg.json` and AST `ast.txt`
- `<lib name>/report`: The directory of logic synthesis reports produced by the commerical EDA tool.
- `<lib name>/netlist.v`: The post-mapping netlist under technology library `<lib name>`.
- `<lib name>/netlist.sdf`: The Standard Delay Format (SDF) file of post-mapping netlist, indicating the wire connections and timing models of cells.
- `<lib name>/netlist.aig`: The And-Inverter Graph (AIG) file of circuit implementations, which is converted from post-mapping netlist with `strash` command in abc Brayton & Mishchenko (2010).

Original Repo: *signal_pipeline*

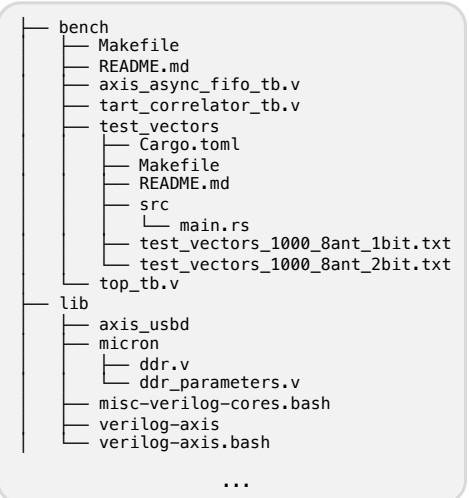

Our Dataset: *signal_pipeline*

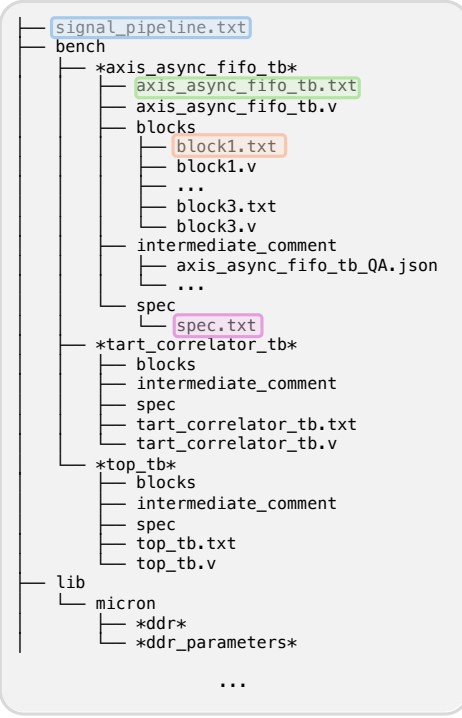

**Repo-Level Annotation**

The repository named "signal_pipeline" is dedicated to handling and processing radio-frequency signals, primarily targeted for applications like software-defined radios and radio transmission systems. It includes a comprehensive suite of modules designed for various aspects of signal processing, such as buffering, correlating, accumulating, and multiplexing of signal data. Key modules like 'toy_correlator', 'tart_correlator', 'correlate', 'visaccum', and 'sigbuffer' function to streamline the data flow through sequence control, data handling, and complex calculations over different clock domains. These are well-integrated via testbench modules ('toy_correlator_tb', 'tart_correlator_tb', etc.) that simulate module operations under varying conditions to ensure reliability and correctness. Additionally, the repository architectures interfaces, including SPI and ULPI for USB, and implements FIFO methodologies (like 'axis_async_fifo') to manage data queues efficiently. Modules such as 'top' and 'controller' manage the overarching hardware connections and operational synchronicity. The design complexity is further managed through implementation on FPGA platforms, with references to specific hardware configurations like GOWIN FPGAs. The repository not only handles signal processing tasks but also simulates the peripheral scenarios via modules like 'radio_dummy' and test configurations ensuring robust module verification and data integrity checks across the system.

**Module-Level Annotation**

The 'axis_async_fifo_tb' is a testbench module for testing an AXI Stream FIFO (First-In, First-Out) interface with different parameter configurations like depth, data width, etc., and conditions like asynchronous resets, data generation, and data readiness. It includes two simulated clock domains for source (s_clk) and master (m_clk), along with internal signals for test inputs and capturing responses. It employs an instance of 'axis_async_fifo' module, stimulating it with generated data, and observes response status signals for validating the main module's operations.

**Block-Level Annotation**

Line number:
[84, 137]
Comment:
This block of code implements an AXIS asynchronous FIFO(First-In, First-Out) buffer. It allows data transfer between two clock domains, with the depth and data width controlled by the parametrized 'DEPTH' and 'DATA_WIDTH'. Special features such as 'KEEP', 'LAST', 'ID', 'DEST', and 'USER' are enabled by their corresponding '_ENABLE' parameters. The AXIS FIFO instance is instantiated with a read clock (m_clk) and write clock (s_clk), and includes ports for data input (s_axis_tdata) and output (m_axis_tdata). Also, there are separate ports to monitor the status of the FIFO buffer, including overflow or the validity of frames.

**Detailed Specification**

Module name: axis_async_fifo_tb.
Module specification: The 'axis_async_fifo_tb' module serves as a testbench for the asynchronous FIFO memory queue which is designed using the AXI (Advanced eXtensible Interface) stream interfaces, specifically referred to as 'axis_async_fifo'. The crucial functionalities include the generation of various testing conditions such as asynchronous resets, data generation, validity, readiness signals, and more.
Input ports incorporate s_clk (source clock), async_rst (asynchronous reset signal), s_axis_tdata (source AXI Stream input data), s_axis_tkeep and s_axis_tvalid (AXI Stream validity signals for tdata), s_axis_tlast (source AXI Stream signal indicating the last transfer in a packet) ......

Figure 2: Illustration of the dataset repository structure with multi-level annotations

Original Repo: *Image-Processor*

```
├── rtl
│   ├── README.md
│   ├── filelist.f
│   ├── process.v
│   └── image.v
```

Our Dataset: *Image-Processor*

```
├── rtl
│   ├── README.md
│   ├── filelist.f
│   ├── process.v
│   └── image.v
├── graph
│   ├── cdfg.json
│   └── ast.txt
├── sky130
│   └── report
│       ├── area.rpt
│       ├── design.rpt
│       ├── power.rpt
│       └── timing.rpt
│   ├── netlist.v
│   ├── netlist.sdf
│   └── netlist.aig
├── asap7
│   └── report
│       ├── area.rpt
│       ├── design.rpt
│       ├── power.rpt
│       └── timing.rpt
│   ├── netlist.v
│   ├── netlist.sdf
│   └── netlist.aig
          ...
```

Snippet of RTL Design (`process.v`)

```verilog
always @(*) begin
  case (state)
  // IDLE state
  IDLE: begin
    mirror_done = 0;
    out_we = 0;
    row_index = 0;
    col_index = 0;
    next_state = Read_first_pixel;
  end
  ... ...
  endcase
end
```

Area Report (`area.rpt`)

```
Library(s) Used: sky130_fd_sc_hd__tt_025C_1v80
Number of ports:                         62
Number of nets:                      201016
Number of cells:                     200978
Number of combinational cells:       102674
Number of sequential cells:           98304
Number of macros/black boxes:             0
Number of buf/inv:                    31778
Number of references:                    13
Combinational area:           611959.402199
Buf/Inv area:                 119281.897054
Noncombinational area:       2951951.062500
Macro/Black Box area:              0.000000
Net Interconnect area:    undefined  (Wire load has zero net area)
Total cell area:              3563910.464699
Total area:                       undefined
```

Snippet of Post-Mapping Netlist (`netlist.v`)

```
sky130_fd_sc_hd__edfxtp_1 \data_reg[23][2][5] ( .D(n90603), .DE(n110473),
.CLK(n80384), .Q(\data[23][2][5] ) );
sky130_fd_sc_hd__edfxtp_1 \data_reg[23][2][4] ( .D(n90179), .DE(n110473),
.CLK(n80384), .Q(\data[23][2][4] ) );
sky130_fd_sc_hd__edfxtp_1 \data_reg[23][2][3] ( .D(n89755), .DE(n110473),
.CLK(n80384), .Q(\data[23][2][3] ) );
sky130_fd_sc_hd__edfxtp_1 \data_reg[23][2][2] ( .D(n89331), .DE(n110473),
.CLK(n80384), .Q(\data[23][2][2] ) );
sky130_fd_sc_hd__edfxtp_1 \data_reg[23][2][1] ( .D(n88907), .DE(n110473),
.CLK(n80383), .Q(\data[23][2][1] ) );
```

Figure 3: Illustration of the dataset repository structure with Circuit Transformation

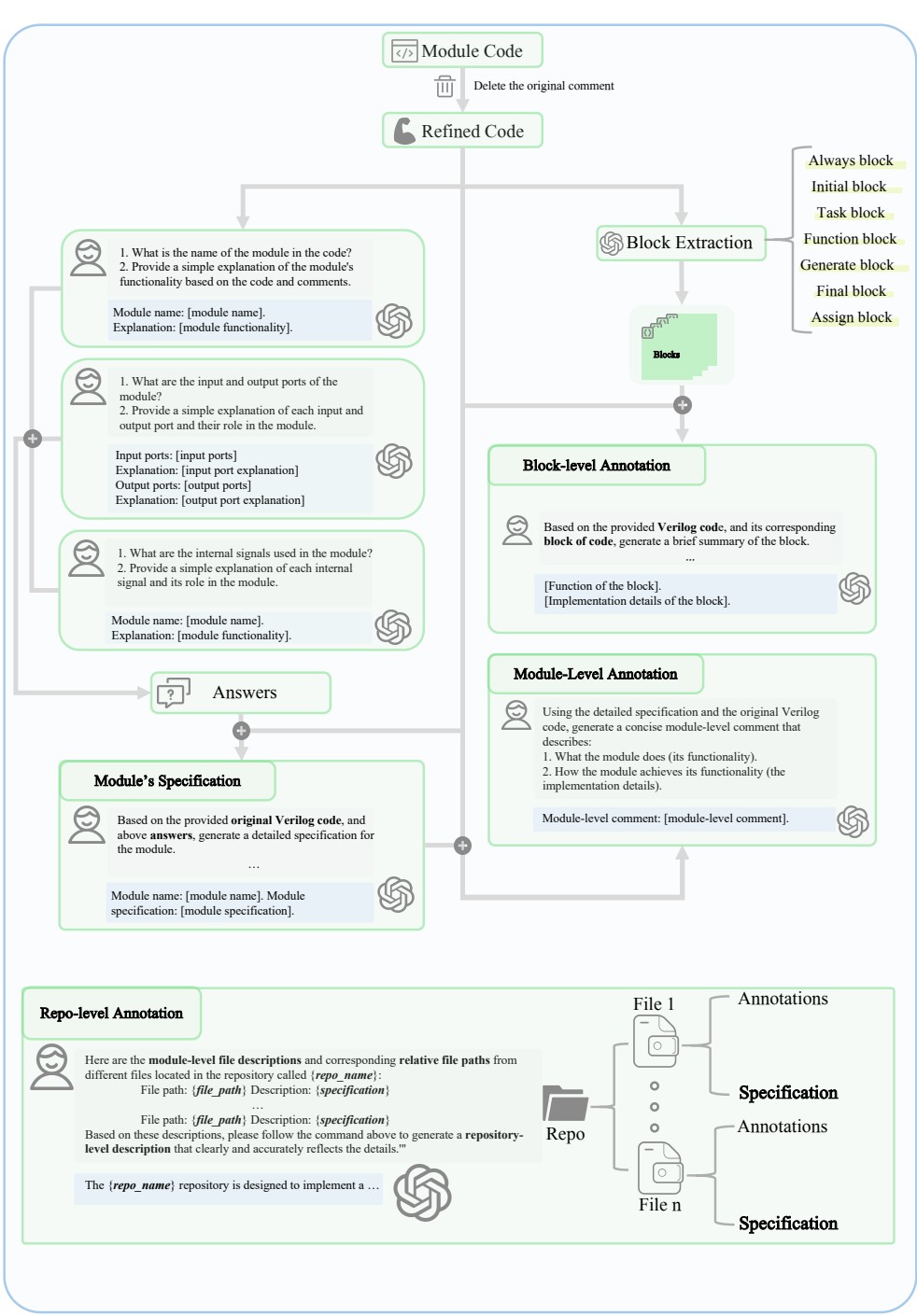

Figure 4: Detailed procedures in our proposed method: CoT code-annotation.

