# OpenReview forum: "DeepCircuitX: Repository-Level RTL Dataset for Code Understanding, Generation, and Multimodal Analysis"
_ICLR.cc/2025/Conference — ICLR 2025 Conference Withdrawn Submission_

### Official Review · Reviewer_H4Ct · 2024-10-27

**Soundness:** 3
**Presentation:** 2
**Contribution:** 2
**Rating:** 3
**Confidence:** 5

**Summary:**

The paper introduces DeepCircuitX, a comprehensive multimodal dataset designed to advance RTL code understanding, generation, and completion tasks in hardware design automation. DeepCircuitX spans repository, file, module, and block-level RTL code, enriched with CoT annotations for detailed functionality and structure descriptions. It includes synthesized netlists and PPA metrics, enabling early-stage design exploration and PPA prediction directly from RTL code. The dataset also establishes benchmarks using models like CodeLlama, CodeT5+, and CodeGen, which may provide a critical resource for advancing RTL code tasks in hardware design automation.

**Strengths:**

1. Hardware design automation is a highly important field. I believe that promoting the development of this field by building large-scale and high-quality datasets is very worthwhile.
2. The dataset presented in this paper is highly diverse, including natural language, code, and kinds of multimodal transformation like AST, CDFG, and netlist.
3. The idea of organizing the dataset at repository, file, module, and block levels is reasonable and aligns with practical hardware design.

**Weaknesses:**

1. The main contribution is not clear. Does this paper propose a higher-quality training dataset or a benchmark? If it contributes a training dataset, the paper should compare training results with previous work (e.g., RTLCoder, BetterV, CodeV, AutoVCoder [1-4]) to demonstrate the advantages of this dataset. Additionally, it should report the model training results using its own dataset in the PPA, rather than only showing results from others who used this dataset. If its main contribution is the benchmark, the paper should provide a more detailed comparison of this benchmark with existing ones, offering more reliable evaluation standards, detailing the composition of the dataset, and conducting a thorough measurement of the performance of existing Verilog models on this benchmark. Specifically, in the code generation task, how does the benchmark ensure correspondence between natural language and code? How is the correctness of code generation tested? Are BLEU, ROUGE, and METEOR good evaluation metrics for the code understanding task?

2. The human evaluation in Section 4.2 indicates that the quality of this dataset still needs improvement, especially in terms of accuracy. This suggests that unresolved problems may exist when using this dataset as either a training set or a benchmark. Besides, how is this result compared with baseline datasets?

3. The concepts of "Chip, IP, Module, and RISC-V" and "module-level,block-level, repository-level" are somewhat confusing. Could the author further explain the relationship between them? Or can the authors provide some concrete examples of them?

[1] RTLCoder: Outperforming GPT-3.5 in Design RTL Generation with Our Open-Source Dataset and Lightweight Solution

[2] BetterV: Controlled Verilog Generation with Discriminative Guidance

[3] CodeV: Empowering LLMs for Verilog Generation through Multi-Level Summarization

[4] AutoVCoder: A Systematic Framework for Automated Verilog Code Generation using LLMs

**Questions:**

See most questions in the weaknesses.
1. Why didn't the authors choose some Verilog code models like RTLCoder for the experiment?
2. Why did the authors use DeepSeek-Coder-V2-lite for the code understanding task but not for the code generation task?
3. A typo in line 338.

---

### Official Review · Reviewer_9Pyv · 2024-10-29

**Soundness:** 2
**Presentation:** 3
**Contribution:** 2
**Rating:** 3
**Confidence:** 4

**Summary:**

The authors introduced an innovative large-scale hardware design dataset to enhance the understanding, generation, and completion ability of large language models for hardware design. The dataset includes a wide range of complete hardware designs at various levels. The author used LLMs to generate multi-level annotations for these files and obtained PPA data using EDA tools. Experiments designed by the author demonstrate that this dataset significantly improves the capabilities of LLMs and other models in hardware design.

**Strengths:**

The author collected a large and comprehensive dataset of complete RTL designs, organized by different levels such as Block, Module, File, and Repo. Using CoT, the author leveraged LLMs to annotate hardware designs hierarchically, introducing alignment between natural language and RTL code at multiple levels. Through significant effort, the author obtained a well-organized dataset that aligns RTL code, natural language descriptions, and PPA metrics. This work is crucial for improving the LLM's hardware design capabilities.

**Weaknesses:**

I really appreciate the author's efforts in dataset collection and organization. However, I think there are still some areas where the article could be improved.

1. LLM-based methods struggle to ensure high-quality alignment

   The DeepCircuitX dataset consists of two parts, obtained using different methods: natural language annotation data and PPA data. For the former, the authors use GPT-4 to generate summaries at different levels. However, VerilogEval[1] has shown that there is a gap between manually written descriptions and those generated by GPT. LLM's performance on VerilogEval-machine surpasses that on VerilogEval-human. Data fully generated by GPT may have a limited impact on improving LLM's effectiveness in real-world applications. The authors evaluated the effectiveness of the generated descriptions through human engineers, but if they could further explore whether GPT-generated descriptions align with human-written descriptions, this work would be more comprehensive.

2. Insufficient details on dataset specifics.

   The authors present statistics for all collected data in Table 1 and the annotated data, which is smaller, in Table 2. In Section 5, the authors state that they selected "high-quality" data for annotation. It would be better if the authors could further explain the criteria for this selection. Additionally, perhaps it was something I missed, but I didn't find any mention of the quantity of PPA data. Most open-source repositories cannot be synthesized easily, and their synthesis setups might differ. I'm curious about the number of designs included in the PPA analysis.

3. The experimental results do not prove the effectiveness of the dataset.

   The authors conducted several experiments to illustrate the effectiveness of the dataset, but there are still areas in the experimental setups that could be improved. (1) The authors evaluated the improvement in LLM on code understanding tasks after fine-tuning on their dataset. I may have missed it, but the authors don't seem to mention how they partitioned the training and test sets. If the models are trained and tested on the same dataset, the results might be less meaningful, especially when the evaluation metrics are based on text similarity. (2) In Table 6, the authors evaluated the improvement in LLM performance on RTLLM and VerilogEval after fine-tuning on their dataset. However, the authors did not specify whether they decontaminated the two test sets from training data. Additionally, The results in Table 6 do not reach the SOTA level [2-3], possibly indicating limited annotation quality in the dataset. (3) The PPA prediction task did not use LLMs for prediction, making this part seem somewhat inconsistent from the rest of the paper. I agree with the authors that PPA awareness is very important for LLMs in hardware design. If the authors could further evaluate whether LLMs can learn PPA-related knowledge, this section would be more interesting.

[1] VerilogEval: Evaluating Large Language Models for Verilog Code Generation

[2] RTLCoder: Outperforming GPT-3.5 in Design RTL Generation with Our Open-Source Dataset and Lightweight Solution

[3] CraftRTL: High-quality Synthetic Data Generation for Verilog Code Models with Correct-by-Construction Non-Textual Representations and Targeted Code Repair

**Questions:**

Please refer to the Weakness section.

---

### Official Review · Reviewer_Dyb6 · 2024-11-03

**Soundness:** 2
**Presentation:** 3
**Contribution:** 2
**Rating:** 5
**Confidence:** 5

**Summary:**

The authors main contribution claims are the following:
A. Introduction of a comprehensive dataset containing over 4,000 repository-level RTL projects, covering chip-level, IP-level, module-level, and RISCV designs, with diverse functional and algorithmic keywords, as well as High-Level Synthesis (HLS) data converted to RTL.

B. Organization of the dataset into four abstract levels as: repository, file, module, and block—allowing for model training at different scales and enabling a wide range of applications in EDA tasks such as synthesis, netlist generation, PPA analysis, and layout design.

C. Development of a Chain of Thought (CoT) annotation method using GPT-4 and Claude to generate detailed comments, descriptions, and question-answer pairs, enhancing training data quality for LLMs in RTL code understanding and generation.

D. Creation of pre-training and evaluation benchmarks for LLMs on RTL-related tasks, including code understanding, completion, and neural network-based PPA prediction, demonstrating the proposed dataset's effectiveness, adaptability, and generalization capabilities for RTL code comprehension and various EDA applications.

**Strengths:**

These work focuses on certain limitations as discussed, integrating AI techniques into the EDA workflow, especially for RTL tasks, offers a promising path to improve current EDA tools for complex chip designs and reduce time-to-market. However, existing RTL datasets have significant limitations: (a) Most datasets are narrowly scoped, often focused on a limited set of circuit types (e.g., processors) or indiscriminately include Verilog files without ensuring correctness or diversity. This lack of variety restricts the generalizability and effectiveness of these datasets. (b) A critical gap is the absence of circuit implementation details, such as netlists and layouts, in RTL datasets. This omission neglects the link between RTL code and its actual circuit implementation, making it challenging to train RTL models that ensure both functional correctness and optimization for Power, Performance, and Area (PPA). Additionally, RTL datasets are often sensitive and privacy-preserved, limiting their accessibility and further restricting diversity.

The authors aimed to address these challenges by creating a comprehensive RTL dataset and demonstrating its effectiveness, adaptability, and generalization capabilities for code comprehension and various EDA applications, supporting model training at multiple scales. The research manuscript is well-written and well-organized overall. The authors also benchmarked several LLMs, comparing both baseline and fine-tuned versions on their proposed datasets, demonstrating that the fine-tuned models outperformed the baseline on several metrics.

**Weaknesses:**

The benchmarking methodology is relatively weak and could be strengthened:

(a) RTL Code Understanding: Table 5 presents the performance of various baseline and fine-tuned LLMs on the proposed RTL code dataset, evaluated using BLEU-4, METEOR, ROUGE-1, ROUGE-2, and ROUGE-L metrics. Please extend this analysis to include other benchmark datasets (should be finetuned on these datasets separately or collectively as well), as discussed in Section 2 (RELATED WORK), to enable a fair comparative analysis and to better highlight the advantages of the proposed dataset over previous datasets.

b) Fine-Tuning on Benchmark Datasets: Please clarify why you chose to fine-tune only on your dataset and benchmark against RTLLM and VerilogEval. Were the baseline models already fine-tuned on these two datasets? If not, please fine-tune the models on additional benchmark datasets as advised in point (a), and compare the metrics against fine-tuning on your proposed dataset. This comparison should be reflected in Table 6. Additionally, please provide citations for RTLLM [Liu, Shang et al., 2024] and VerilogEval [Liu, Mingjie et al., 2023].

(c) PPA Prediction: If feasible, please extend this approach to PPA prediction, as shown in Table 7.

Finally, please discuss a recent relevant work, “Dehaerne, E., Dey, B., Halder, S., & De Gendt, S. (2023). A deep learning framework for Verilog autocompletion towards design and verification automation. arXiv preprint arXiv:2304.13840,” which includes a recent benchmarking dataset on Verilog.

**Questions:**

This has already been addressed in the Weaknesses section.

**Details Of Ethics Concerns:**

Have the authors cited or referenced all repositories and relevant websites from which they obtained code, datasets, and RTL projects/files? This should be verified, along with adherence to any applicable license agreements.

---

### Official Review · Reviewer_2uHs · 2024-11-05

**Soundness:** 2
**Presentation:** 3
**Contribution:** 2
**Rating:** 3
**Confidence:** 5

**Summary:**

This paper introduces a dataset named DeepCircuitX to advance the development of hardware design automation. DeepCircuitX includes original RTL code, synthesized netlists, natural language descriptions, PPA metrics, and AST/CDFG graphs. The field of hardware design automation indeed needs a high-quality, open-source dataset. Although the paper has a meaningful motivation, it unfortunately fails to prove the quality of the proposed DeepCircuitX dataset, and the dataset lacks a multimodal format in the experiments.

**Strengths:**

The paper has a meaningful motivation, and the writing is easy to understand.

**Weaknesses:**

**Motivation and Contribution**

1. The first contribution mentions HLS, but it is never discussed later in the paper.
2. It seems the dataset is only for LLMs. Other AI research in EDA fields could also benefit from a high-quality open-source dataset, but this is not discussed.

**Datasets**

3. The proposed dataset is described as “multimodal,” but what modalities does DeepCircuitX include besides text? Does multimodality mean that AST and CDFG are graph formats like PNG or something else?
4. PPA should be reported by an open-source tool like Yosys. DeepCircuitX uses DC to report PPA, which may introduce difficulties for future in-depth research.

**Experiments**

5. The experiments in this paper do not adequately demonstrate the quality of the proposed DeepCircuitX dataset.

**Questions:**

**Dataset**

1. Why is RISCV a separate level from chip-level, IP-level, and module-level?
2. Why not use AST to divide RTL code into distinct functional blocks? Regular expressions and LLMs are not good parsers.
3. In Table 2, what is the difference between module-level and repo-level descriptions for a module-level design? Why do RISCV and module-level designs lack block-level descriptions?
4. For RTL code completion, completing from incomplete code that contains more than just the module header seems more meaningful. Why is only the module header typically provided?
5. Does each design contain multi-level natural language descriptions and synthesized netlists?
6. Does DeepCircuitX separate the original design Verilog and testbench?

**Experiments**

7. For human evaluation, please showcase some annotations of scores 1, 2, 3, and 4. Scores in Table 4 are meaningless without examples showing which annotations score 4 and which score 1, especially since large language models often output misleading content.
8. For RTL code understanding, please provide details of the training and test datasets.
9. For RTL code completion and generation, how many codes (n in pass@k metric) are sampled to calculate pass@k? To my knowledge, pass@k differs from the correctness of the model’s top k outputs.
10. Before training, did DeepCircuitX deduplicate RTLLM and VerilogEval for RTL code completion and generation?
11. In Table 6, code completion and code generation are two tasks according to 3.2.2, but why is only one result presented?
It seems that pass@1 and pass@5 are lower than RTLCoder when fine-tuning CodeLlama. Does this mean DeepCircuitX is lower quality compared to the dataset proposed by RTLCoder?

---

### Note · Authors · 2024-11-19

I have read and agree with the venue's withdrawal policy on behalf of myself and my co-authors.